# L-PRF Secretome from Both Smokers/Nonsmokers Stimulates Angiogenesis and Osteoblast Differentiation In Vitro

**DOI:** 10.3390/biomedicines12040874

**Published:** 2024-04-16

**Authors:** Susana Ríos, Lina Gabriela González, Claudia Gilda Saez, Patricio Cristian Smith, Lina M. Escobar, Constanza Eugenia Martínez

**Affiliations:** 1School of Dentistry, Faculty of Medicine, Pontificia Universidad Catolica de Chile, Santiago 8330024, Chile; susana.rios@uc.cl (S.R.); psmithf@uc.cl (P.C.S.); 2Faculty of Dentistry, Universidad Nacional de Colombia, Bogotá 111321, Colombialmescobarm@unal.edu.co (L.M.E.); 3School of Medicine, Faculty of Medicine, Pontificia Universidad Catolica de Chile, Santiago 8331150, Chile; csaezs@uc.cl; 4Faculty of Dentistry, Universidad de los Andes, Santiago 7620086, Chile

**Keywords:** leukocyte- and platelet-rich fibrin, secretome, cigarette smoking, angiogenesis, osteoblast, cell differentiation

## Abstract

Leukocyte and Platelet-Rich Fibrin (L-PRF) is part of the second generation of platelet-concentrates. L-PRF derived from nonsmokers has been used in surgical procedures, with its beneficial effects in wound healing being proven to stimulate biological activities such as cell proliferation, angiogenesis, and differentiation. Cigarette smoking exerts detrimental effects on tissue healing and is associated with post-surgical complications; however, evidence about the biological effects of L-PRF derived from smokers is limited. This study evaluated the impact of L-PRF secretome (LPRFS) derived from smokers and nonsmokers on angiogenesis and osteoblast differentiation. LPRFS was obtained by submerging L-PRF membranes derived from smokers or nonsmokers in culture media and was used to treat endothelial cells (HUVEC) or SaOs-2 cells. Angiogenesis was evaluated by tubule formation assay, while osteoblast differentiation was observed by alkaline phosphatase and osterix protein levels, as well as in vitro mineralization. LPRFS treatments increased angiogenesis, alkaline phosphatase, and osterix levels. Treatment with 50% of LPRFS derived from smokers and nonsmokers in the presence of osteogenic factors stimulates in vitro mineralization significantly. Nevertheless, differences between LPRFS derived from smokers and nonsmokers were not found. Both LPRFS stimulated angiogenesis and osteoblast differentiation in vitro; however, clinical studies are required to determine the beneficial effect of LPRFS in smokers.

## 1. Introduction

Leukocyte and platelet-rich fibrin (L-PRF) is an autologous second-generation platelet concentrate that stimulates wound healing and regeneration in many clinical situations. Previous reports have demonstrated that L-PRF releases growth factors and biomolecules, keeping a temporary fibrin scaffold at the wound site to stimulate tissue healing and reduce pain and discomfort [1,2].

L-PRF is obtained by the following defined protocol: centrifugation (408× *g*, for 12 min) of the venous blood sample into a silica-coated plastic tube without anticoagulants. L PRF clot enriched in leucocytes and platelets is obtained from the middle portion of the tube, removing the erythrocytes. This L-PRF clot can be compressed to get membranes or plugs at the wound receptor site [3]. There is growing in vitro and in vivo evidence that supports the application of L-PRF derived from nonsmokers to stimulating wound healing during both medical and dental conditions, including the treatment of leg ulcers, periodontal defects, sinus lift surgery, alveolar ridge preservation, and as an adjuvant in gingival graft procedures for gingival recession treatment [2,3,4,5,6]. These beneficial effects may be linked to the impact of L-PRF on cell proliferation, migration, angiogenesis, bone differentiation, and inhibition of osteoclast activity [7,8,9]. Both in vitro and in vivo studies have reported that L-PRF from nonsmokers may stimulate osteoblast proliferation, increasing early and late bone markers. L-PRF has been proposed to potentiate bone neoformation in critical size defects in rat calvaria [10].

On the other hand, scientific evidence supports smoking’s negative impact on all phases of wound healing. Previous studies have reported that cigarette smoking is a highly concentrated source of free radicals and toxic chemical compounds, while smoke reduces tissue perfusion and oxygenation, the bactericidal potential of neutrophils and monocytes, and collagen synthesis and deposition at the wound site [11]. A recent report demonstrated that long-term exposure to nicotine or cigarette smoke condensates to gingival fibroblasts in vitro has a negative effect on cell proliferation, migration, and extracellular matrix deposition, while prompting an increase in inflammatory mediators such as Interleukin 6 and 8 [12]. All these alterations in the cellular biological activities negatively influence the wound healing tissue success. 

Interestingly, although smoking has been demonstrated to exert significant detrimental effects on tissue healing and regeneration [11,13,14], few studies have evaluated the impact of smoking on L-PRF beneficial effects for stimulating wound healing in smokers. Our lab’s previous work has reported similar features for L-PRF derived from smokers compared to nonsmokers. In these studies, we found a comparable release of growth factors, mechanical properties, and stimulation of migration and cell proliferation of human periodontal ligament stromal cells in vitro [15,16]; however, L-PRF biological activities, such as angiogenesis and osteoblast differentiation, which are indispensable to establishing bone regeneration have not been evaluated in L-PRF derived from smokers. Therefore, this study aimed to assess the effects of L-PRF secretome derived from smokers and nonsmokers in angiogenesis and osteoblast differentiation in vitro. 

## 2. Materials and Methods

### 2.1. L-PRF Sampling and L-PRF Secretome Recovery

Venous blood samples were obtained from four healthy nonsmokers and four smoker volunteers who smoked at least ten cigarettes daily. All procedures were approved by the Ethical Scientific Committee from the Pontificia Universidad Catolica de Chile (ID 170706007). Blood was collected in 10 mL silica-coated plastic tubes (red cap Becton Dickinson., Franklin Lakes, NJ, USA) and centrifuged at 408× *g* (2700 RPM) for 12 min [17]. L-PRF clots were isolated from the middle of the tubes, and erythrocytes were eliminated. L-PRF fibrin clots were compressed using a stainless-steel box (Xpression Box, Biohorizons, Birmingham, AL, USA) to get membranes. Two membranes were obtained per donor, and the exudates released during compression were used to evaluate cotinine levels. Each L-PRF membrane was submerged in Dulbecco’s Modified Eagle’s Medium Low Glucose (DMEM, HyClone GE Life Sciences; Marlborough, MA, USA) in a cell culture plastic dish at 37 °C for 24 h. After this incubation, L-PRF-conditioned medium, referred to from this point as L-PRF secretome (LPRFS), was recovered and kept at −80° until its use [15]. 

### 2.2. Enzyme-Linked Immunosorbent Assay (ELISA)

Cotinine levels were measured in the exudates and L-PRF secretome (LPRFS) from smoker/nonsmoker donors to confirm the smoking habit using a cotinine ELISA kit (ab285286, Abcam, Waltham MA, USA). Samples were incubated and read at 450 nm. PDGF-BB (DBB00), FGF-2 (DFB50), and IL-6 (D6050) protein levels were analyzed by ELISA (all from R&D Systems; McKinley Place NE, MN, USA) in the L-PRF secretomes. For IL-6 positive control, primary human gingival fibroblasts derived from 8 donors were starved for 24 h, and the conditioned medium was recovered and analyzed. Briefly, LPRS or conditioned medium from gingival fibroblasts were incubated following manufacturer instructions and read at 450 nm. Data was analyzed using the GraphPad Prism 10.0.2 software (Boston, MA, USA). All samples were evaluated in triplicate.

### 2.3. Tube Formation In Vitro

Angiogenesis in vitro was evaluated according to Arnaoutova and Kleinman [18]. Human umbilical vein endothelial cells (HUVEC) pre-screened for angiogenesis were acquired in Lonza (CX.C2519A, Lonza, Basel, Switzerland). For tubule formation in vitro, 15,000 cells per well were seeded in 96-well cell culture plates over a cultrex base membrane extract reduced in growth factors (CULTREX (RD 3433-005-01, R&D Systems, McKinley Place NE, MN, USA)). Cells were incubated with LPRFS from nonsmokers or smokers at 50% and diluted in an endothelial basal medium for 6 h. As a positive control, HUVEC cells were incubated with an endothelial growth medium (Basal medium plus supplements according to manufacturer indications (Endothelial cell growth medium kit-2 C-22111: 2% Fetal calf serum, EGF 5 ng/mL, bFGF 10 ng/mL, IGF 20 ng/mL, VEGF 0.5 ng/mL, ascorbic acid one µg/mL, heparine 22.5 µg /mL and 0.2 µg /mL of hydrocortisone), Promocell, Heidelberg, Germany) supplemented with 35 ng/mL of bFGF (SIGMA−Aldrich, Burlington, MA, USA). The negative controls used were an endothelial basal medium (EBM) or endothelial growth medium supplemented with 35 ng/mL of bFGF and Sulforaphane 15 µM (SIGMA−Aldrich, Burlington, MA, USA), an angiogenesis inhibitor. Finally, cells were incubated with calcein 6.25 µg/mL (Invitrogen, Carlsbad, CA, USA) and observed in an inverted epifluorescence microscope at 4×. Images were taken and quantified by Wimasis (Wim Tube, Onimagin Technologies, Córdoba, Spain) to determine tube formation. Data were analyzed by GraphPad prism. All experiments were performed in triplicate. 

### 2.4. Western Blotting

The Osteosarcoma SaoS-2 cell line was obtained from the European Collection of Authenticated Cell Cultures (89050205, Lonza, Basel, Switzerland) and seeded in 6-plate cell cultures at 200,000 cells/per well in a conventional cell culture medium (DMEM plus 10% FBS). After 24 h of serum starvation, cells were incubated with LPRS derived from smokers or nonsmokers at 100 or 50% concentration and with or without osteogenic factors (10 mM β-glycerophosphate, 0.1 μM dexamethasone and 50 μg/mL ascorbic acid 2-phosphate. Merck, Rahway, NJ, USA) for 48 h. As a control, cells were incubated with DMEM plus FBS 10% with or without osteogenic factors. After treatments, total proteins were extracted with a lysis buffer, including a proteases inhibitor cocktail (539131, Calbiochem, San Diego, CA, USA), and the protein content was quantified with a commercial kit (23227, Thermo Fischer Scientific, Waltham, MA, USA). SDS-PAGE electrophoresis was performed using 4–20% precast polyacrylamide gels (Mini protean, TGX-Stain free, Biorad, Hercules, CA, USA) and 20–40 ug of protein per lane. A protein ladder (spectra multicolor protein ladder (26634, Thermo Fischer Scientific, Waltham, CA, USA) was included in each gel. Proteins were transferred to polyvinylidene difluoride membranes (Thermo Fischer Scientific, Waltham, CA, USA) and incubated overnight at 4 °C with primary antibodies, including anti-osterix (1:5000 (R&D Systems; McKinley Place NE, MN, USA)), anti-alkaline phosphatase (1:5000, Abcam, Waltham MA, USA), and, as housekeeping was used, an anti-GAPDH rabbit antibody (1:20,000, SIGMA−Aldrich, Burlington, MA, USA). Then, membranes were incubated with HRP-conjugated secondary antibodies (1:10,000, SIGMA−Aldrich, Burlington, MA, USA) for two hours at room temperature. The membranes were incubated with an enhanced chemiluminescence detection kit (ECL Pierce or SuperSignal West Femto, Thermo Fischer Scientific, Waltham, MA, USA) and exposed to a Molecular Imager (ImageQuant LA500, General Electric, Piscataway, NJ, USA). Protein relative levels were quantified by ImageJ 1.54g software [19]. 

### 2.5. Osteogenic Differentiation 

For in vitro bone differentiation, samples of the Osteosarcoma SaoS-2 cell-line were kept in DMEM supplemented with 10% fetal bovine serum (Gibco, Thermo Fischer Scientific, Waltham, MA, USA) plus antibiotic-antimycotic solution (100 U/mL penicillin, 100 μg/mL streptomycin, 0.25 μg/mL amphotericin B, SIGMA−Aldrich, Burlington, MA, USA). For bone differentiation, cells were seeded and incubated for 11 days with DMEM supplemented with FBS 10% plus an osteogenic supplement (10 mM β-glycerophosphate, 0.1 μM dexamethasone and 50 μg/mL ascorbic acid 2-phosphate. Merck, Rahway, NJ, USA) as the positive control. Additionally, Saos-2 cells were incubated with LPRFS derived from smokers/nonsmokers at 100% or 50% concentrations and diluted in DMEM both with or without osteogenic supplements. 

### 2.6. Alizarin Red Stain

SaoS-2 cells were treated in an osteogenic medium for 11 days. Cells were fixed in paraformaldehyde (SIGMA−Aldrich, Burlington, MA, USA) 4% for 15 min at room temperature. Then, cells were washed with phosphate-buffered saline and incubated with Alizarin Red (SIGMA−Aldrich, Burlington, MA, USA) 40 mM for 30 min. Cells were rinsed five times with ultrapure water to eliminate unbonded stains. Images of each well were taken, and calcium deposits were quantified using ImageJ software [20].

### 2.7. Immunostaining

Saos-2 cells were grown on coverslips treated for 48 h with LPRS derived from smokers and nonsmokers. Then, they were fixed with paraformaldehyde 4%, blocked with 1% albumin, and incubated overnight with mouse monoclonal anti-osterix (1:1500; (MAB7547, R&D Systems; McKinley Place NE, MN, USA)). Subsequently, cells were incubated with a secondary antibody Alexa fluor 555(1:400, Invitrogen, Carlsbad, CA, USA) and counterstained with DAPI (1:5000, Invitrogen, Carlsbad, CA, USA), as previously reported. Coverslips were observed in an epifluorescence microscope, and representative images were taken. 

### 2.8. Statistical Analyses

All experiments were conducted in triplicate. For normality data analysis, the Shapiro–Wilk test was done. After confirming normality distribution, one-way-ANOVA or student-*t* parametric tests were used with Tukey’s post hoc test to analyze multiple comparisons. On the other hand, the Kuskall−Wallis test and Dunn’s multiple comparisons test were conducted for data without normal distribution. Data was analyzed using GraphPad Prism software (GraphPad, Boston, MA, USA).

## 3. Results 

### 3.1. LPRFS from Smokers and Nonsmokers Releases Similar Levels of PDGF-BB and FGF-b Growth Factors

First, all exudates derived from L-PRF clot compression to get membranes were analyzed for cotinine as a smoking reliable marker. L-PRF Exudates derived from smokers showed significantly increased cotinine levels compared to exudates obtained from nonsmokers (Figure 1A). At the same, LPRF secretome derived only from smokers (LPRFS-S) evidenced low levels of cotinine after 24 h of LPRFS recovery. To evaluate the presence of proteins related to angiogenesis and bone differentiation, protein levels of FGF-2 and PDGF-BB were analyzed in LPRFS. Elevated levels of PDGF-BB were detected in LPRFS from both smokers and nonsmokers without differences between the groups (Figure 1B). Likewise, FGF-2 levels did not show statistical differences between LPRFS derived from smokers/nonsmokers (Figure 1C). On the other hand, Interleukin-6 levels were almost undetectable in both LPRFS-S and LPRFS-NS compared to the high levels observed in conditioned medium derived from starved human primary gingival fibroblasts (Figure 1D).

### 3.2. LPRFS Derived from Smokers and Nonsmokers Promotes Angiogenesis In Vitro

Angiogenesis is a crucial step for successful wound healing and bone formation; therefore, the effects of LPRFS in angiogenesis were evaluated in an in vitro model. Human endothelial umbilical cord cells (HUVEC) were seeded over a basal membrane extract low in growth factors, treated for 6 h with LPRFS-S or LPRFS-NS diluted in endothelial basal medium 1:1, and stained with calcein to visualize cell morphology and tube formation. Treatment with both LPRFS-S and LPRFS-NS significantly stimulates tubule formation in vitro compared to negative controls (basal medium or the angiogenic medium supplemented with sulforaphane, an angiogenesis inhibitor). Cells treated with LPRFS-S/LPRFS-NS significantly increased the total tube number (Figure 2D), total tube length (Figure 2E), mean loop area (Figure 2G), and mean loop perimeter (Figure 2H). HUVEC cells established connections to form capillary-like tubes only in the presence of LPRFS-S or LPRFS-NS or the positive control with angiogenic factors (Figure 2A,B) compared to the negative controls (Figure 2C).

### 3.3. LPRFS Derived from Smokers/Nonsmokers Increases Bone Differentiation In Vitro

SaOs-2 cells were incubated with LPRFS derived from smokers and nonsmokers at 100% or 50% concentration and with or without osteogenic factors for 48 h in vitro. Afterward, cells were lysed, and immunoblotting for alkaline phosphatase and Osterix was performed. LPRFS treatment at both concentrations (100 or 50%) and derived from smokers and nonsmokers stimulates the relative levels of alkaline phosphatase in a similar way to controls (DMEM + FBS 10% with or without osteogenic factors) (Figure 3A–J). Significant differences were observed between alkaline phosphatase protein levels in cells treated with LPRFS-S at 50% plus osteogenic factors compared to controls (DMEM + FBS 10% with or without osteogenic factors) or LPRFS-S or LPRFS-NS at 50%. (Figure 3A–C). 

Likewise, Osterix (Osx), a transcription factor involved in osteoblast commitment and differentiation, was evaluated by immunoblotting and immunostaining early in the in vitro bone differentiation process. SaOs-2 cells were treated for 48 h with the conventional cell culture medium with or without osteogenic factors, and Osx protein levels remained unchanged (Figure 3A,D–F). However, significant differences in Osx protein levels were observed in cells treated with LPRFS derived from smokers and nonsmokers (LPRFS-S and LPRFS-NS) at both 100% and 50% of LPRFS concentration in the presence of osteogenic factors compared to controls (Figure 3A,D–J). However, differences between LPRFS-S and LPRFS-NS were not detected.

To evaluate the effects of LPRFS treatment on the mineralization in vitro, SaOS-2 cells were incubated with LPRFS-S or LPRFS-NS with or without osteogenic factors for 11 days. Then, calcium deposits were evidenced with the Alizarin red stain (Figure 4). In the presence of osteogenic factors, LPRFS and FBS treatments stimulated the calcium deposits (Figure 4A–D). However, significant differences were found in cells treated with LPRFS-S and LPRFS-NS at 50% concentration in the presence of osteogenic factors compared to the conventional cell culture medium supplemented with osteogenic factors. (Figure 4A–D). Calcium deposits in the cells incubated with LPRFS-S or LPRFS-NS at 100% concentration plus osteogenic factors were very similar to the control (FBS + osteogenic factors). However, differences between cells incubated with LPRFS-S or LPRFS-NS were not detected (Figure 4A–D).

## 4. Discussion

This study demonstrated that L-PRF secretome (LPRFS) derived from L-PRF membranes from smokers and nonsmokers similarly stimulates angiogenesis and bone differentiation in vitro. L-PRF is a source of growth factors and biomolecules for stimulating tissue wound healing in vitro and in vivo [3,10]; however, most previously reported evidence includes L-PRF derived from nonsmoker healthy donors. L-PRF has been proven effective in treating periodontal defects and sinus lift procedures evaluated through clinical and radiographic parameters. This evidence involving only healthy nonsmokers and histological evidence of bone regeneration using L-PRF is limited [4]. According to the last best evidence consensus statement of the American Academy of Periodontology related to the use of biologicals in clinical procedures, platelet-derived fractions, mainly L-PRF, can promote soft tissue healing and bone formation [4]. They could be beneficial in situations where diseases or habits such as smoking are associated with possible poor outcomes; however, comparative studies and a detailed characterization of L-PRF derived from those individuals are necessary [4] Nevertheless, to our knowledge, few studies have assessed whether LPRF from smoker subjects may exert a different effect on tissue healing. Tobacco smoke is a complex and reactive mixture containing approximately 5000 chemical compounds. This mixture is probably the most significant source of toxic chemical exposure and chemically mediated disease in humans [21]. It is widely known that smoking is associated with a higher risk of developing periodontitis; additionally, it affects the outcomes of active periodontal therapies in a dose-dependent manner, is a risk factor for bone loss associated with peri-implantitis and orthopedic implant loosening, and negatively affects bone metabolism [22,23].

Smoking has also been associated with post-surgical complications such as flap failure, hematoma, wound dehiscence, surgical site infection, and impaired wound healing after head and neck reconstructive surgery [11,24]. As we said before, L-PRF has been considered to be an autologous source of biomolecules and an alternative for stimulating tissue wound healing. Still, limited evidence related to L-PRF from smokers has been reported. Previous reports of our lab comparing L-PRF from smokers and nonsmokers found similar mechanical properties, biomolecule release profiles, and proliferation and migration in periodontal ligament stromal cells [15,16]. These biological activities are indispensable for successful wound healing and tissue regeneration.

On the other hand, bone formation is a complex process directed by a molecular program of transcription factors and proteins involving the new formation of blood vessels or angiogenesis [10]. This study evaluated the effects of LPRFS recovered from smokers and nonsmokers in angiogenesis and bone differentiation in vitro. After smoking confirmation by cotinine marker in the L-PRF exudates (released after L-PRF clot compression to get L-PRF membranes), we observed similar levels of PDGF-BB and FGF-2 in the LPRFS derived from smokers and nonsmokers. These results were similar to FGF-2 and PDGF protein levels reported previously by Ye et al. in crevicular gingival fluid in smokers [25]. Both growth factors have been related to increased osteoblast proliferation and differentiation for new bone formation [26,27]. Interleukin 6 (Il-6) is a well-proinflammatory cytokine that has been associated with increased serum levels in smokers [28]; however, almost indetectable protein levels of this cytokine were observed in LPRFS from smokers and nonsmokers. A limitation of this study was the number of biomolecules analyzed in LPRFS. More studies are required to evaluate a more detailed cytokine composition in LPRFS and possible biomolecules associated with cigarette smoke’s adverse health effects in the L-PRF membranes. 

Then, we found that LPRFS derived from smokers and nonsmokers stimulated angiogenesis similarly. Previous reports have demonstrated that the L-PRF secretome and L-PRF membranes derived from nonsmokers could promote tubule formation in vitro and in vivo [29,30]. However, to our knowledge, this is the first study analyzing the angiogenic effect of LPRFS derived from smokers. Future studies are needed to understand the molecular mechanisms associated with this effect of LPRFS derived from smokers and nonsmokers on tubule formation.

Subsequently, the protein levels of the alkaline phosphatase and Osterix (Osx) transcription factor, both essential biomolecules during early bone differentiation, were evaluated after 48 h of LPRFS treatment derived from nonsmokers and smokers with or without osteogenic factors. LPRFS treatments stimulated alkaline phosphatase protein levels similarly to controls in both LPRFS-S and LPRFS-NS. In cells treated with LPRFS derived from nonsmokers, increased alkaline phosphatase levels were observed, as in previous reports [7,31]. Nevertheless, there are no other studies involving LPRFS derived from smokers. 

Osx is a transcription factor expressed during bone differentiation and involved in osteoblast differentiation, maturation, and activity. Osx is engaged in complex communications among different bone cells and plays a role in the bone microenvironment. At the same time, Osx induces the expression of mature osteoblast genes such as collagen type-I a1 (Col1a1), Osteonectin, Osteopontin, Osteocalcin, and Bone sialoprotein (BSP), which are all necessary for productive osteoblasts during bone formation [32]. In this study, LPRFS from smokers and nonsmokers stimulates the expression of osx protein levels. However, a significant reduction in the Osx protein levels was observed in cells treated with LPRFS derived from smokers and nonsmokers at 100% and 50% in the presence of osteogenic factors compared to controls. Evidence of the L-PRF effects on the Osx expression is scarce [7]. A previous report using periodontal ligament cells observed stimulation of mRNA osx levels after three days of treatment with 20% LPRFS with or without osteogenic factors in a similar way to control [33]. This result differs from our study (probably due to different experimental conditions), where we evaluated 100 or 50% LPRFS concentration and a different time point of evaluation. The reduction in Osx protein levels in our study could be explained by the Osx-associated degradation of proteasome-dependent by E3 Ligase reported previously [34]; however, the exact mechanism that regulates Osx stability and degradation is unknown. Regardless, in order to observe a reduction in the OSX protein levels in early osteoblast differentiation process, LPRFS from smokers and nonsmokers significantly stimulated the mineralization in vitro. 

Similarly to our study, recently, Hoshikawa et al. reported that in vitro models of bone differentiation such as MC3T3 and UCBTERT-21 cells have evidenced high Osterix protein levels after 12 or 24 h of Bone morphogenetic protein-4 (BMP-4) treatment, a growth factor strongly associated with osteoinduction and a reduction after 48 h in the presence of osteogenic factors [34]. Nevertheless, a limitation of our study was that it did not include more evaluation points because the LPRFS quantity derived from human donors was limited to verify calcium deposits after 11 days of treatment with three medium changes per week and a significant number of milliliters of LPRFS to treat cells for the Western blot experiments. However, preliminary assays were performed to verify the LPRFS significant stimulation of Osx protein levels after 24 h. 

Finally, after osteoblast differentiation, calcium deposits were evaluated after 11 days of LPRFS treatment in the presence of osteogenic factors. Significant calcium deposits were observed in SaOS-2 cells treated with 50% LPRFS derived from smokers/nonsmokers in the presence of osteogenic factors compared with the conventional medium. Similar results were reported by Li et al. using periodontal ligament cells and You et al. in MG-63 osteosarcoma-derived cells after LPRFS treatment [31,33]. However, these studies did not include LPRFS derived from smokers. In this study, we demonstrated the stimulation of calcium deposits in SaOs-2 cells after LPRFS treatment derived from smokers and nonsmokers. Nevertheless, more studies are needed to clarify the molecular mechanisms associated with the beneficial effects of LPRFS in osteoblast differentiation. 

## 5. Conclusions

In conclusion, with the limitations of these experimental in vitro analyses, results suggest that LPRFS derived from smokers/nonsmokers were able to stimulate angiogenesis, increase the protein levels of alkaline phosphatase, and increase the calcium deposits in vitro. However, in vivo and clinical studies are indispensable in determining the effects of LPRFS derived from smokers as an alternative to stimulating wound healing and regeneration in this population. 

## Figures and Tables

**Figure 1 biomedicines-12-00874-f001:**
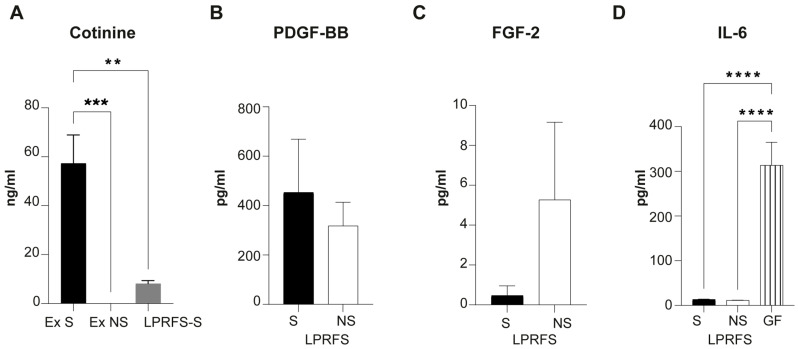
LPRFS from smokers and nonsmokers have similar PDGF-BB and FGF-2 growth factors and IL-6 levels. L-PRF membranes were obtained from S and NS after L-PRF Clot compression, and exudates released were recovered for cotinine quantification. L-PRF membranes were submerged in DMEM for 24 h to obtain LPRFS. (**A**) Cotinine was determined in L-PRF exudates derived from S, NS, and LPRFS-S. Quantification of (**B**) PDGF-BB, (**C**) FGF-2, and (**D**) IL-6 levels in LPRFS and conditioned medium of starved human gingival fibroblast as IL-6 control. ** *p* ≤ 0.0022; *** *p* ≤ 0.002; **** *p* ≤ 0.0001 Ex = L-PRF exudates, S = smokers, NS = nonsmokers, LPRFS = L-PRF secretome.

**Figure 2 biomedicines-12-00874-f002:**
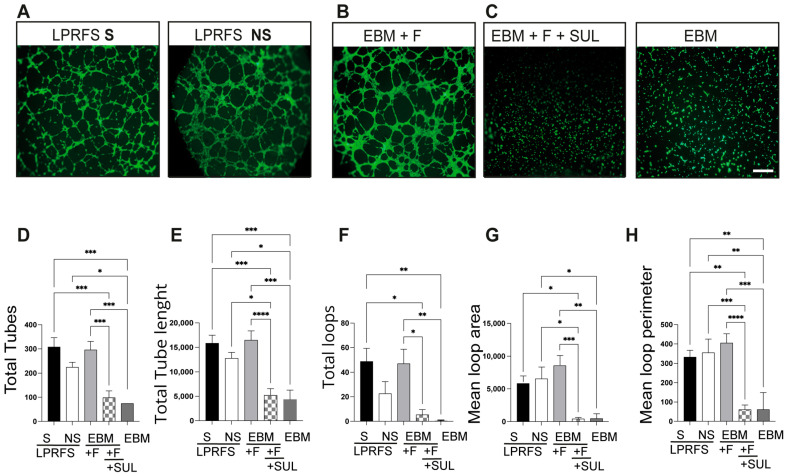
LPRFS from smokers and nonsmokers stimulates angiogenesis in vitro. HUVEC cells were treated for 6 h with 50% LPRFS-S or LPRFS-NS or positive (EBM + FGF) and negative (EBM + FGF + SUL or EBM) controls. Cells were stained with calcein and observed in an epifluorescence inverted microscope at 4×. Representative images of live stain HUVEC cells treated with (**A**) LPRFS or (**B**) Positive and (**C**) negative Controls. Quantification of (**D**) total tubes, (**E**) total tube length, (**F**) total loop area, (**G**) mean loop area, and (**H**) mean loop perimeter. * *p* ≤ 0.0140; ** *p* ≤ 0.085; *** *p* ≤ 0.005; **** *p* ≤ 0.001. S = smokers, NS = nonsmokers, LPRFS = L-PRF secretome, EBM = endothelial basal medium, F = FGF = fibroblast growth factor, SUL = sulphoraphane. Scale Bar = 50 microns.

**Figure 3 biomedicines-12-00874-f003:**
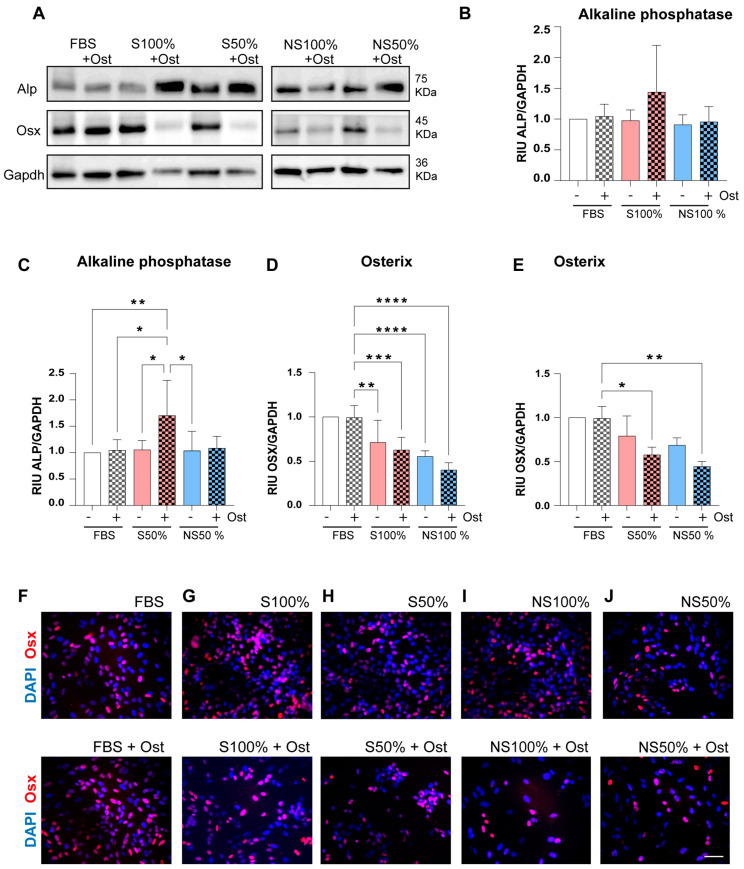
LPRFS derived from smokers and nonsmokers modulates Alkaline phosphatase and Osterix protein levels. SaOs-2 cells were treated for 48 h with LPRFS-S or LPRFS-NS with or without osteogenic factors. DMEM supplemented with FBS 10% with or without osteogenic factors was used as a control. (**A**) Representative images of a Western blot for Alkaline phosphatase (Alp), Osterix (Osx), and Glyceraldehyde-3-phosphate dehydrogenase (Gapdh) in SaOs-2 cells treated in indicated conditions. Relative quantification of (**B**) Alp protein levels in cells treated with 100% LPRFS, (**C**) Alp protein levels in cells treated with 50% LPRFS, (**D**) Osx protein levels in cells treated with 100% LPRFS and (**E**) Osx protein levels in cells treated with 50% LPRFS. Representative images of Osx immunostaining in red and DAPI nuclear stain in blue, SaOs-2 cells were treated as indicated in (**F**) Controls, (**G**) 100% LPRFS-S (**H**) 50% LPRFS-S, (**I**) 100% LPRFS-NS, (**J**) 50% LPRFS-NS with or without osteogenic factors. * *p* ≤ 0.0455; ** *p* ≤ 0.0074; *** *p* ≤ 0.003; **** *p* ≤ 0.001. S = smokers, NS = nonsmokers, LPRFS = L-PRF secretome, Ost = osteogenic factors, RIU = relative intensity units. Scale Bar = 50 microns.

**Figure 4 biomedicines-12-00874-f004:**
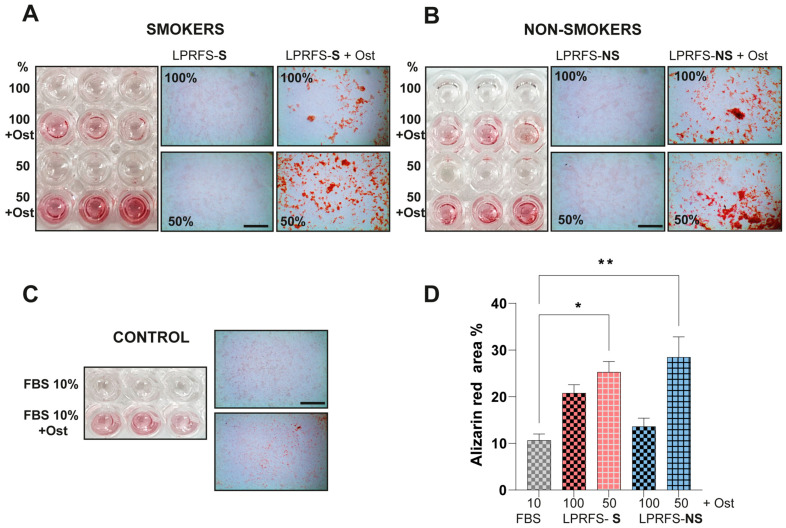
LPRFS derived from smokers and nonsmokers stimulates mineralization in vitro. SaOs-2 cells were treated for 11 days with LPRFS-S or LPRFS-NS with or without osteogenic factors. DMEM supplemented with FBS 10% with or without osteogenic factors was used as a control. Cells were stained with alizarin red stain. Representative images of cells treated with (**A**) LPRFS-S, (**B**) LPRFS-NS, or (**C**) 10% FBS. (**D**) Quantification of percentage area stained with alizarin red. * *p* ≤ 0.0247; ** *p* ≤ 0.061. S = smokers, NS = nonsmokers, LPRFS = L-PRF secretome, Ost = osteogenic factors. Scale Bar = 500 microns.

## Data Availability

The data presented in this study are available on request from the corresponding authors.

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
