# Peer review of "L-PRF Secretome from Both Smokers/Nonsmokers Stimulates Angiogenesis and Osteoblast Differentiation In Vitro"

_biomedicines, 2024, doi:10.3390/biomedicines12040874_

Round 1

Reviewer 1 Report

Comments and Suggestions for Authors

1. This is an observational study, which is more descriptive without mechanistic insights.

2. The quality of all figures is too low, which cannot support the acceptance of the journal.

3. A scale bar should be added in the Figure 2A-C.

The image in Figure 4 in the current form of this manuscript is blurry, which is not acceptable for publication in scientific journals. Moreover, additional evidence for the molecular mechanisms by which the L-PRF Secretome-induced an increase in angiogenesis and osteoblast differentiation is required for this study.

Author Response

Thank you for allowing us to submit a revised draft of our manuscript. We are grateful to the reviewers for their insightful comments on this paper. We have incorporated changes to reflect most of their suggestions and highlighted the changes in manuscript version 2.

  1. This is an observational study, which is more descriptive without mechanistic insights.

Thanks to the reviewer for pointing out this issue. We agree with the reviewer that our study is the first step in identifying the effects of smoking on L-PRF abilities to induce angiogenesis and osteoblast differentiation. Our results suggest that FGF and EGF in LPRF secretome from smokers and nonsmokers stimulate tubule formation in HUVEC cells in vitro. However, the mechanisms associated are unknown and need to be established as the next step in this study. As in osteoblast differentiation, more analysis of the intracellular pathway’s interactions is required to increase the in vitro mineralization. In lines 312 and 362, we have included the need to establish future studies to understand the molecular mechanisms associated with our first approximation to both biological activities in the discussion section.

  1. The quality of all figures is too low, which cannot support the acceptance of the journal.

Thanks a lot for your suggestion. We apologize for the low quality of the figures sent in this first manuscript version. All figures are now high quality, allowing us to visualize the results adequately. 

  1. A scale bar should be added in the Figure 2A-C.                                      Thanks, a scale bar was added to the figure 2A-C.

4. The image in Figure 4 in the current form of this manuscript is blurry, which is not acceptable for publication in scientific journals. Moreover, additional evidence for the molecular mechanisms by which the L-PRF Secretome-induced an increase in angiogenesis and osteoblast differentiation is required for this study

Thanks a lot for your consideration. All the figures, including figure 4, were improved in quality. We agree with the reviewer that determining the molecular mechanisms beneath the LPRFS effects on angiogenesis and osteoblast differentiation is important. However, our study was the first step to observing and establishing the positive effects of LPRFS in smokers and nonsmokers.

Reviewer 2 Report

Comments and Suggestions for Authors

The article entitled “L-PRF Secretome from Smokers/Nonsmokers Stimulates 2 Angiogenesis and Osteoblast Differentiation In Vitro” is interesting and brings new information to the literature. Only a few adjustments are necessary prior to publication.

1.     The abstract is out of the journal’s guidelines. The subheadings should not appear and the abstract should have no more than 200 words. Please, adjust.

2.     The introduction section lacks background regarding the utility of L-PRF.

3.     The authors opted to separate smokers from non-smokers and in the introduction section no sufficient background was provided. Is quite clear this segregation.

4.     Figure 3 lacks quality. Please, adjust.

5.     The conclusion section should not have a limitation statement, please, include it in the discussion section.

Author Response

Thank you for allowing us to submit a revised draft of our manuscript. We are grateful to the reviewers for their insightful comments on this paper. We have incorporated changes to reflect most of their suggestions and highlighted the changes in manuscript version 2.

  1. The abstract is out of the journal’s guidelines. The subheadings should not appear and the abstract should have no more than 200 words. Please, adjust.

We apologize for not following the instructions for authors in the abstract. Thanks a lot for your suggestion. The abstract was re-written, and it has 200 words without subheadings.

  1. The introduction section lacks background regarding the utility of L-PRF.

Thanks a lot for pointing out this issue. The introduction included a phrase clarifying the effects of l-PRF on stimulating wound healing in many clinical situations. All changes were highlighted in yellow.

  1. The authors opted to separate smokers from non-smokers and in the introduction section no sufficient background was provided. Is quite clear this segregation.

Thank you. We have clarified the introduction section. We decided to segregate both L-PRF from smokers and nonsmokers because a lot of evidence previously reported supporting the use of L-PRF in surgical procedures in medicine and dentistry is based on L-PRF derived from healthy non-smokers. A gap in evidence exists related to the effects of smoking on L-PRF properties to stimulate wound healing in smokers. Smokers have more possibilities of having post-surgical complications such as infection and dehiscence due to this habit. Therefore, we analyzed and compared both smokers and nonsmokers separately.

       4. Figure 3 lacks quality. Please, adjust.

Thanks a lot for your suggestion. We apologize for the low quality of the figures sent in this first manuscript version. All figures are now high-quality, allowing us to adequately illustrate the results. 

  1. The conclusion section should not have a limitation statement, please, include it in the discussion section.

Thanks a lot for your consideration. The conclusion subheading was eliminated, and the conclusion was incorporated into the final paragraph of the discussion section.

Reviewer 3 Report

Comments and Suggestions for Authors

The manuscript by Rios et al. focuses on the interesting topic of stimulating angiogenesis and osteogenesis in vitro by L-PRF-derived secretome. The secretome was obtained from two kinds of samples - L-PRF membraines obtained from non-smoking and smoking volunteers. The manuscript is interested in general and the study was designed and performed properly. Some issues have to be corrected to improve the quality of paper.

Minor remarks:

1. Please consider changing the title - if there are no differences between smokers and non-smokers it should be highlighted, or at least indicate that "L-PRF secretome from both smokers and non-smokers stimulates ....."

2. Introduction must be more informative and give some broader perspective, e.g. on other patophysiological processes influenced by smoking.

3. Materials and methods - line 75 - please correct the centrifugation forces: RCF is in relation to g, so probably it was 700 x g (equal to 408 RPM?)

4. Figure 2 - x-axis captions are hard to decode: what is E+F sample? which control is positive and which is negative?

5. Figure 3 - what is RIU on y axis?

6. Figure 3F - please explain what was the rationale to stain OSX on blots as well as on coverslips?

Comments on the Quality of English Language

English is fine, only some minor corrections are needed.

Author Response

Thank you for allowing us to submit a revised draft of our manuscript. We are grateful to the reviewers for their insightful comments, which improved the quality of this paper. We have incorporated changes to reflect most of their suggestions and highlighted the changes in manuscript version 2.

  1. Please consider changing the title - if there are no differences between smokers and non-smokers it should be highlighted, or at least indicate that "L-PRF secretome from both smokers and non-smokers stimulates ....."

Thanks for your excellent suggestion; we agree, and the title was changed as you suggested.

  1. Introduction must be more informative and give some broader perspective, e.g. on other patophysiological processes influenced by smoking.

Thank you for your suggestion. A brief paragraph summarizing the main pathophysiological changes related to smoking during wound healing was added to the introduction section (lanes 55-64)

  1. Materials and methods - line 75 - please correct the centrifugation forces: RCF is in relation to g, so probably it was 700 x g(equal to 408 RPM?)

Thanks a lot for your consideration. We corrected the information in lane 83.

  1. Figure 2 - x-axis captions are hard to decode: what is E+F sample? which control is positive and which is negative?

Thanks again for your suggestion regarding the quality of the figures. We improved the quality and font size in figure 2 to make visualizing the information on the X and Y axes easy.

        5. Figure 3 - what is RIU on y axis?

Thank you for your question. RIU is related to relative intensity units, which quantify protein levels in western blots.

      6. Figure 3F - please explain what was the rationale to stain OSX on blots as well as on coverslips?

Thanks a lot for pointing out this suggestion. We evaluated the OSX protein levels by immunostaining in the whole cell to visualize the changes observed in the western blot. We considered exploring the same research objective with one or more techniques to reproduce the results as an excellent alternative to verifying their reproducibility.

Round 2

Reviewer 1 Report

Comments and Suggestions for Authors

The reviewer has no further comment.